# Residual Bootstrap Exploration for Stochastic Linear Bandit

**Shuang Wu**[1]        **Chi-Hua Wang**[1,2]        **Yuantong Li**[1]        **Guang Cheng**[1]

[1]Department of Statistics, University of California, Los Angeles, Los Angeles, California, USA
[2]Department of Statistics, Purdue University, West Lafayette, Indiana, USA

## Abstract

We propose a new bootstrap-based online algorithm for stochastic linear bandit problems. The key idea is to adopt residual bootstrap exploration, in which the agent estimates the next step reward by re-sampling the residuals of mean reward estimate. Our algorithm, residual bootstrap exploration for stochastic linear bandit (`LinReBoot`), estimates the linear reward from its re-sampling distribution and pulls the arm with the highest reward estimate. In particular, we contribute a theoretical framework to demystify residual bootstrap-based exploration mechanisms in stochastic linear bandit problems. The key insight is that the strength of bootstrap exploration is based on collaborated optimism between the online-learned model and the re-sampling distribution of residuals. Such observation enables us to show that the proposed `LinReBoot` secure a high-probability $\tilde{O}(d\sqrt{n})$ sub-linear regret under mild conditions. Our experiments support the easy generalizability of the `ReBoot` principle in the various formulations of linear bandit problems and show the significant computational efficiency of `LinReBoot`.

## 1 INTRODUCTION

Stochastic linear bandit is an online learning problem that the learning agent acts by pulling arms, where each arm is associated with a feature vector, then learning the arms information from the corresponding random rewards. In such problems, the typical goal of a learning agent is to maximize its cumulative reward. Learning more about an arm (explore) or pulling the arm with the highest estimated reward (exploit)

leads to the well-known *exploration- exploitation trade-off*, which is the central trade-off captured in many decision-making applications in modern online service industries. Consequently, the design of stochastic linear bandit algorithms demands an easy-generalizable implementation across various contextualize actions and reward generation processes.

In the past decade of bandit literature, such demands have invited researchers to investigate bootstrap-based exploration-exploitation trade-offs and have drawn rising attention [Baransi et al., 2014, Eckles and Kaptein, 2014, Osband and Van Roy, 2015, Vaswani et al., 2018, Hao et al., 2019, Kveton et al., 2019b, Wang et al., 2020]. Yet, prior works on bootstrap-based bandit algorithms focus on provable multi-armed bandit algorithms and only provide a limited empirical evaluation of bootstrap-based stochastic linear bandit algorithms, and their theoretical counterpart remains unknown. Such knowledge gap of bootstrapping stochastic linear bandit persuades our investigation on the provable bootstrap-based stochastic linear bandits: **Can we theoretically and empirically support the validity and easy-generalizability of bootstrapping procedure in stochastic linear bandit algorithms design?** In particular, we aim to deliver a generic framework to demystify the bootstrap optimism in stochastic linear bandit problems and validate the easy generalizability of the bootstrap principle across various contextual linear bandit problems.

**Contributions.** We introduce `LinReBoot` algorithms that implement Residual Bootstrap Exploration for stochastic linear bandit problem with sub-linear regret. We theoretically show that `LinReBoot` secures $\tilde{O}(d\sqrt{n})$ regret where $d$ is the dimension of features. This sub-linear regret bound matches the regret bound of the same order as those theoretical results of Linear Thompson Sampling algorithms. The key to achieving such sub-linear regret guarantee is to carefully manage and collaborate sample and bootstrap optimism (Section

*Accepted for the 38th Conference on Uncertainty in Artificial Intelligence* (UAI 2022).

4.1). In particular, by measuring the "sample-bootstrap optimistic estimated discrepancy ratio" of the optimal arm, `LinReboot` successfully avoids over or under exploration and theoretically secures sub-linear mean regret with high-probability. To our knowledge, this is the first theoretical analysis to support the validity and efficiency of the residual bootstrap-based procedure for stochastic linear bandit problems. We empirically show that `LinReBoot` rivals or exceeds competing algorithms including Linear Thompson Sampling, Linear PHE, Linear GIRO, and Linear UCB under stochastic linear bandit problem as well as more complicated linear bandit settings. These significant results support the easy-generalizability of proposed `LinReBoot`. In summary, our contributions are as follows:

- Propose `LinReBoot` algorithms that implement Residual Bootstrap Exploration in linear bandit problems without boundness assumption of rewards.
- Theoretically show that `LinReBoot` secures $\tilde{O}(d\sqrt{n})$ regret, matching the regret bound of the same order as those theoretical results of Linear Thompson Sampling algorithms.
- Empirically show that `LinReBoot` rivals or exceeds baseline algorithms and supports that `LinReBoot` is easy-generalizable among linear bandit problems.

**Related Works.** Bootstrap-based contextual bandit algorithms design has been actively studied in the last half-decade and drawn a surge of interest from both theoretical studies and industrial practice [Elmachtoub et al., 2017, Eckles and Kaptein, 2014, Osband et al., 2016, Kveton et al., 2019b, Hao et al., 2019]. Bootstrap-based bandit algorithm design is a paradigm of sequential decision-making based on an exploration mechanism with no pre-defined mean reward model. Such paradigm enjoys a decisive advantage that engineers are free to deploy any reward model of interests without painful adaption to problem structure [Kveton et al., 2019b,a]. `ReBoot` [Wang et al., 2020] provided a theoretical logarithmic regret guarantee for multi-armed bandit (MAB) and empirical investigation to validate the easy generalizability of the `ReBoot` principle. Our work aims to provide a theoretical guarantee for the bootstrap-based linear bandit algorithms and empirically investigate more general contextual linear bandit setting to validate the `ReBoot` principle.

One close related work is [Kveton et al., 2020a] which introduces perturbation of past samples for exploration under stochastic linear bandit problem. The limitation of [Kveton et al., 2020a] is the boundness of rewards, indicating many broader classes of rewards such as Gaussian rewards are not applicable with a theoretical guarantee. In contrast, the proposed `LinReBoot` algorithms relax the boundness reward assumption and thus validate bootstrap-based bandit algorithms in wider

bandit environments with a broader class of reward generation processes.

Early works about exploration in bandit problems [Abbasi-Yadkori et al., 2011, Langford and Zhang, 2007, Dani et al., 2008] are practical but no guarantee of the optimality. Some works [Wang et al., 2020, Kveton et al., 2019b,a, Thompson, 1933, Auer et al., 2002] provide well designed exploration for bandit problems and have their own principles for adopting to more general problems. In these works, three principles including `ReBoot` [Wang et al., 2020], `GIRO` [Kveton et al., 2019b] and `PHE` [Kveton et al., 2019a] are devising exploration mechanism based on up-to-now history instead of on pre-defined reward model in the other two principles `TS` [Thompson, 1933] and `UCB` [Auer et al., 2002]. Our work generalizes `ReBoot` into stochastic linear bandit problems.

**Notations.** Let $[n]$ be set $\{1, 2, ..., n\}$. $\mathbf{1}$ is a vector with all ones and $\boldsymbol{I}$ is the identity matrix. For a vector $\boldsymbol{v}$, $\|\boldsymbol{v}\|_2$ is 2-norm of $\boldsymbol{v}$ and $\|\boldsymbol{v}\|_{\boldsymbol{A}}^2 := \sqrt{\boldsymbol{v}^\top \boldsymbol{A} \boldsymbol{v}}$ for a semidefinite matrix $\boldsymbol{A}$. Let $\langle \cdot, \cdot \rangle$ be the inner product operation. Denote $\mathcal{F}_t$ as the history of randomness up to round $t$. $\mathbb{E}_t[\cdot] := \mathbb{E}[\cdot | \mathcal{F}_{t-1}]$ is defined as the conditional expectation given $\mathcal{F}_{t-1}$ and $\mathbb{P}_t(\cdot) := \mathbb{P}(\cdot | \mathcal{F}_{t-1})$ is defined as the conditional probability given $\mathcal{F}_{t-1}$. $\mathbb{I}\{\cdot\}$ is indicator function. For a set or event $E$, we denote its complement as $\bar{E}$. $N(\mu, \sigma^2)$ is Gaussian distribution with mean $\mu$ and variance $\sigma^2$. We use $\tilde{O}$ for big $O$ notation up to logarithmic factor.

## 2 STOCHASTIC LINEAR BANDIT

**Contextualize Action Set.** In stochastic linear bandit problem, we identify the actions with $d-$dimensional features from $\mathcal{A} \subset \mathbb{R}^d$ and assume $|\mathcal{A}|$, the size of the action set, is finite. Let $K := |\mathcal{A}|$ be the number of actions (arms), $\boldsymbol{x}_k \in \mathbb{R}^d$ be the context vector of the $k$-th arm, that is, $\mathcal{A} = \{\boldsymbol{x}_1, ..., \boldsymbol{x}_K\}$.

**Reward generating mechanism.** The reward function is parameterized by $\boldsymbol{\theta} \in \mathbb{R}^d$ such that, at time $t$ the agent chooses an action $I_t \in [K]$ with feature $X_t = \boldsymbol{x}_{I_t} \in \mathcal{A}$, the reward is generated by

$$Y_t \equiv \langle X_t, \boldsymbol{\theta} \rangle + \epsilon_t. \tag{1}$$

Specifically, the reward obtained by the agent at round $t$ when pulling arm $I_t = k$ is generated from a distribution with mean $\mu_k := \boldsymbol{x}_k^\top \boldsymbol{\theta}$, conditioning on context $\boldsymbol{x}_k$. The property of noise $\epsilon_t$ is described in Assumption 2. Furthermore, denote the recieved reward by $r_{I_t}$ and the reward random variable by $Y_t$ at round $t$.

**Regret.** Without loss of generality, assume that arm 1 is the unique optimal arm, that is $\mu_1 > \mu_k \ \forall k \neq 1$.

The optimal gap of the $k$-th arm is $\Delta_k := \mu_1 - \mu_k \geq 0$. The expected $n$-round regret is denoted as

$$R_n := \sum_{k=2}^{K} \Delta_k \mathbb{E}[\sum_{t=1}^{n} \mathbb{I}\{I_t = k\}]. \qquad (2)$$

The goal of the agent is to maximize the expected cumulative reward in $n$ rounds, which is equivalent to minimizing the expected regret $R_n$.

**Assumption 1.** *(Boundness assumptions) True parameter $\boldsymbol{\theta}$ is bounded: $\|\boldsymbol{\theta}\|_2 \leq S_2$.*

Besides, we denote $L$ as the upper bound for context vectors: $\|\boldsymbol{x}_k\|_2 \leq L$ for all $k \in [K]$. Assumption 1 is referred to the boundness assumptions in the stochastic linear bandit literature and is to ensure the regret is bounded if the agent pulls any sub-optimal actions (see Section 5 in [Abbasi-Yadkori et al., 2011]).

**Assumption 2.** *(Noise Clipping assumption) Noise process $\{\epsilon_t\}_{t=1}^{\infty}$ described in (1) satisfies that for some $L_1, L_2 > 0$,*

$$e^{L_1 \eta^2} \leq \mathbb{E}[e^{\eta \epsilon_t} | \mathcal{F}_{t-1}] \leq e^{L_2 \eta^2}, \ \forall \eta \geq 0, \qquad (3)$$

*where $\mathcal{F}_{t-1} = \{\epsilon_1, I_1, \cdots, \epsilon_{t-1}, I_{t-1}\}$.*

Assumption 2 implies that stochastic process $\{\epsilon_t\}_{t=1}^{\infty}$ is conditionally sub-gaussian with constant $L_2$. $L_1$ contributes to the lower bound of moment generating function suggested by [Zhang and Zhou, 2020]. Note that the Assumption 2 allows heteroscedasticity among different arms by choosing $L_2$ as the largest variance among arms. Such heteroscedasticity consideration arises and has been identified as a challenge in applications of Bayesian optimization [Kirschner, 2021, Cowen-Rivers et al., 2020].

# 3 RESIDUAL BOOTSTRAP EXPLORATION

## 3.1 REBOOT PRINCIPLE

This section presents essential proof of concepts to implement `ReBoot` principle [Wang et al., 2020]. In general, each round of interaction, the decision policy admits four subroutines to implement `ReBoot` principle: 1) Learning, 2) Fitting, 3) Bootstrapping, and 4) Exploring. Following elaborates on each subroutine:

**1) Model Learning.** The first subroutine outputs a learned model based on current collected data. Our implementation learns the parameter $\boldsymbol{\theta}$ in Eq.(1) by some user-specified model.

**2) Data Fitting.** The second subroutine fits the current data set with the learned model in the previous

subroutine and then outputs the residual set. Intuitively, the residuals measure the *goodness of fit* of the learned model and should drop a hint on the right amount of exploration. In other words, the residuals should suggest a right magnitude of exploration bonus in decision policy (8). How to manage and integrate uncertainty behind residuals into the exploration mechanism of policy is the main challenge.

**3) Residuals Bootstraping.** The third subroutine associates the residuals obtained the last subroutine with a bootstrapping distribution. Instead of maintaining a belief distribution on a parameter in the Bayesian approach, `ReBoot` principle maintains a bootstrapping distribution on the statistical error based on residuals. The challenge is to justify the efficacy of residual-based optimism construction in both theory and practice.

**4) Actions Exploring.** The fourth subroutines sample the exploration bonus from the bootstrapping distribution and output an index for each action. Such bootstrap procedure is more computationally efficient than prior efforts since this procedure only requires drawing a sample from the bootstrapping distribution. The challenge is to prove that such bootstrap procedure secures sub-linear regret in theory.

## 3.2 LINREBOOT ALGORITHM

We propose the Linear Residual Bootstrap Exploration algorithm (`LinReBoot`, Algorithm 1) for stochastic linear bandit problems. This section elaborates the four subroutines in Section 3.1 for the proposed `LinReBoot`.

**1)** `LinReBoot` uses ridge regression procedure, whose learned parameter is $\hat{\boldsymbol{\theta}}_t$ (4b) and estimated mean reward for arm $k$ is $\hat{\mu}_{k,t}$ (4c). Such way to estimate mean reward is easy to manage the confidence [Abbasi-Yadkori et al., 2011]. Thus, we focus on confidence management for the bootstrap-based exploration.

**Ridge Regression Procedure.** `LinReBoot` fits linear model at round $t$ as follow,

$$\boldsymbol{V}_t = \boldsymbol{X}_{t-1}^{\top} \boldsymbol{X}_{t-1} + \lambda \boldsymbol{I}, \qquad (4a)$$

$$\hat{\boldsymbol{\theta}}_t = \boldsymbol{V}_t^{-1} \boldsymbol{X}_{t-1}^{\top} \boldsymbol{Y}_{t-1}, \qquad (4b)$$

$$\hat{\mu}_{k,t} = \boldsymbol{x}_k^{\top} \hat{\boldsymbol{\theta}}_t, \ \forall k \in [K], \qquad (4c)$$

where $\boldsymbol{X}_{t-1} = (X_1, ..., X_{t-1})^{\top} \in \mathbb{R}^{(t-1) \times d}$. The $\tau$-th row of $\boldsymbol{X}_{t-1}$ is the context $X_{\tau}^{\top}$ for $\tau \in [t-1]$, $\boldsymbol{Y}_{t-1} = (Y_1, ..., Y_{t-1})^{\top}$ is reward vector whose elements are rewards up to round $t-1$. $\lambda$ denotes the regularization level. $\boldsymbol{V}_t$ denotes the sample covariance matrix up to round $t$ and $\hat{\boldsymbol{\theta}}_t$ is the ridge estimation of target parameter $\boldsymbol{\theta}$ in (1). $\hat{\mu}_{k,t}$ denotes the estimated mean of arm $k$ based on history. Note that the first $K$ rounds in proposed `LinReBoot` is fully exploring each arm

once. In other words, $I_t = t$ when $t \in [K]$, indicating $\boldsymbol{X}_K := (\boldsymbol{x}_1, ..., \boldsymbol{x}_K)^\top \in \mathbb{R}^{K \times d}$. We call this $\boldsymbol{X}_K$ the context matrix with rank $r \leq \min(K, d)$ and singular values $\sigma_1, ..., \sigma_r$. Also define $\sigma_{\min}^2 \leq \sigma_i^2 \leq \sigma_{\max}^2$, $\forall i \in [r]$. With these definitions, we make a mild assumption about the shrinkage effect of ridge regression:

**Assumption 3.** *(Validity of Ridge Regression) The singular value decomposition of context matrix $\boldsymbol{X}_K$ is denoted as $\boldsymbol{X}_K := \boldsymbol{G\Sigma U}$ where $\boldsymbol{G} \in \mathbb{R}^{K \times K}$, $\boldsymbol{\Sigma} \in \mathbb{R}^{K \times d}$ and $\boldsymbol{U} \in \mathbb{R}^{d \times d}$. Define $\boldsymbol{\Omega} := \boldsymbol{\Sigma}(\boldsymbol{\Sigma}^\top \boldsymbol{\Sigma} + \lambda \boldsymbol{I})^{-1} \boldsymbol{\Sigma}^\top \in \mathbb{R}^{K \times K}$ and $\boldsymbol{Z} := \boldsymbol{G\Omega\Sigma U} \in \mathbb{R}^{K \times d}$. Let $\boldsymbol{z}_1 \in \mathbb{R}^d$ be the first row of $\boldsymbol{Z}$. Given any $\lambda > 0$, there exists a corresponding positive scalar $S_1$ such that $|\boldsymbol{x}_1^\top \boldsymbol{\theta} - \boldsymbol{z}_1^\top \boldsymbol{\theta}| \geq S_1$ for the $\theta$ in (1).*

**Remark 1.** *Assumption 3 provides a lower bound of the absolute difference between true mean $\boldsymbol{x}_1^\top \boldsymbol{\theta}$ and normalized mean $\boldsymbol{z}_1^\top \boldsymbol{\theta}$ of the optimal arm. Note that if $\lambda \to 0$, then $\boldsymbol{z}_1 \to \boldsymbol{x}_1$ and $S_1 \to 0$. Thus this scalar $S_1$ measures the small perturbation on the mean of the optimal arm when the ridge regression procedure is applied. This $\boldsymbol{Z}$ can be interpreted as a ridge shrinkage context matrix [Goldstein and Smith, 1974]. One important phenomenon of online ridge regression is that even if the ridge estimator is biased, the shrinkage effect from ridge estimation provides exploration for the agent leading to making a correct decision. The positive scalar $S_1$ describes the shrinkage effect on the context. That is, the existence of $S_1$ indicates the ridge procedure is valid and its shrinkage effect exists.*

**2)** The fitting part of `LinReBoot` outputs the residuals under the linear model framework,

$$e_{k,t,i} = r_{k,i} - \hat{\mu}_{k,t}, \; \forall i \in [s_{k,t-1}], \qquad (5)$$

where $s_{k,t-1} := \sum_{\tau=1}^{t-1} \mathbb{I}\{I_\tau = k\}$ is the number of times pulling arm $k$ by round $t-1$, $r_{k,i}$ is the $i$-th reward of arm $k$ by round $t-1$. The *goodness of fit* of the learned ridge regression model can be summarised by Residual Sum of Squares(RSS) [Archdeacon, 1994] which is defined as

$$RSS_{k,t} := \sum_{i=1}^{s_{k,t-1}} e_{k,t,i}^2. \qquad (6)$$

Such measure plays an important role in the residual bootstrap exploration mechanism.

**3)** The third part is Residuals Bootstrapping. This subroutine is independent of the model which suggests the power of generalizability of `ReBoot` principle. `ReBoot` principle requires the computation of the exploration bonus [Mammen, 1993], which is $s_{k,t-1}^{-1} \sum_{i=1}^{s_{k,t-1}} \omega_{k,t,i} e_{k,t,i}$, where $\{\omega_{k,t,i}\}_{i=1}^{s_{k,t-1}}$ is residual bootstrap weights for arm $k$ at round $t$.

---

**Algorithm 1** `LinReBoot`

---

**Require:** $\lambda$, $s_{1,0} = ... = s_{K,0} = 0$
  **for** $t = 1, ..., n$ **do**
    **if** $t < K + 1$ **then**
      $I_t \leftarrow t$
    **else**
      $\boldsymbol{V}_t \leftarrow \boldsymbol{X}_{t-1}^\top \boldsymbol{X}_{t-1} + \lambda \boldsymbol{I}$
      $\hat{\boldsymbol{\theta}}_t \leftarrow \boldsymbol{V}_t^{-1} \boldsymbol{X}_{t-1}^\top \boldsymbol{Y}_{t-1}$
      **for** $k = 1, ..., K$ **do**
        $e_{k,t,i} \leftarrow r_{k,i} - \boldsymbol{x}_k^\top \hat{\boldsymbol{\theta}}_t, \; \forall i \in \{s_{k,t-1}\}$
        Generate $\{\omega_{k,t,i}\}_{i=1}^{s_{k,t-1}}$
        $\tilde{\mu}_k \leftarrow \boldsymbol{x}_k^\top \hat{\boldsymbol{\theta}}_t + s_{k,t-1}^{-1} \sum_{i=1}^{s_{k,t-1}} \omega_{k,t,i} e_{k,t,i}$
      **end for**
      $I_t \leftarrow \underset{k \in [K]}{\arg\max} \; \tilde{\mu}_k$
    **end if**
    $s_{I_t,t} \leftarrow s_{I_t,t-1} + 1$ and $s_{k,t} \leftarrow s_{k,t-1}. \; \forall k \neq I_t$
    Pull arm $I_t$ and get reward $r_{I_t, s_{I_t}}$
    $\boldsymbol{X}_t \leftarrow \begin{bmatrix} \boldsymbol{X}_{t-1} \\ \boldsymbol{x}_{I_t}^\top \end{bmatrix}$ and $\boldsymbol{Y}_t \leftarrow \begin{bmatrix} \boldsymbol{Y}_{t-1} \\ r_{I_t, s_{I_t}} \end{bmatrix}$
  **end for**

---

**Choice of Bootstrapping Weights.** The bootstrap weights considered in this work are i.i.d with zero mean and variance $\sigma_\omega^2$. They are independent of the noise process $\{\epsilon_t\}_{i=1}^\infty$. In the literature of bootstrap procedure [Mammen, 1993] , the choices of bootstrap weights distribution include Gaussian weights, Rademacher weights and skew correcting weights. In `LinReBoot`, we adopt the Gaussian bootstrap weights to enable an efficient implement described at section 3.3.

**4)** The last subroutine is the action exploring based on residual bootstrap. More specifically, for arm $k$ at round $t$, `LinReBoot` adds exploration bonus from residual bootstrapping on the estimated mean $\hat{\mu}_{k,t}$ as follow,

$$\tilde{\mu}_{k,t} = \hat{\mu}_{k,t} + \frac{1}{s_{k,t-1}} \sum_{i=1}^{s_{k,t-1}} \omega_{k,t,i} e_{k,t,i}, \qquad (7)$$

then agent pulls arm with the highest bootstrapped mean,

$$I_t \equiv \arg\max_{k \in [K]} \tilde{\mu}_{k,t}. \qquad (8)$$

Note that the variance of bootstrapped mean $\tilde{\mu}_{k,t}$ is $\sigma_\omega^2 s_{k,t-1}^{-2} RSS_{k,t}$, indicating an adaptive amount of extra exploration is controlled by $s_{k,t-1}$ and $RSS_{k,t}$.

**Short Summary.** Our proposed `LinReBoot` has following steps at round $t > K$,

**1)** Ridge estimation: compute $\boldsymbol{V}_t, \hat{\boldsymbol{\theta}}_t$.
**2)** Finding residuals for each arm: for arm $k$, compute $\hat{\mu}_{k,t}$ and $\{e_{k,t,i}\}_{i=1}^{s_{k,t-1}}$.

**3)** Compute Bootstrapped mean for each arm: for arm $k$, generate $\{\omega_{k,t,i}\}_{i=1}^{s_{k,t-1}}$ and compute $\tilde{\mu}_{k,t}$ (7).

**4)** Pull arm with the highest $\tilde{\mu}_{k,t}$ then observe reward.

Algorithm 1 describes `LinReBoot`. The strength of `LinReBoot` is its easy generalizability across different bandit problems including linear bandits and even more complicated structured problems (Appendix D.1).

**Remark 2.** *(`LinTS` perturbs system parameter estimate, `LinReBoot` perturbs expected reward estimates) Compare with the `LinTS` in [Agrawal and Goyal, 2013b], in which `LinTS` samples a perturbed parameter $\tilde{\boldsymbol{\theta}}_t^{LinTS} = \hat{\boldsymbol{\theta}}_t + \beta_t \boldsymbol{V}_t^{-1/2} \boldsymbol{\eta}_t$ with scaling $\beta_t$ and appropriate independent noise $\boldsymbol{\eta}_t$ (defined in [Agrawal and Goyal, 2013b]). Our proposed `LinReBoot` samples a perturbed expected reward $\tilde{\mu}_{k,t}^{LinReBoot} = \langle \hat{\boldsymbol{\theta}}_t, \boldsymbol{x}_k \rangle + \frac{1}{s_{k,t-1}} \sum_{i=1}^{s_{k,t-1}} w_{k,t,i} e_{k,t,i}$. That is, `LinReBoot` is perturbing the expected reward estimate via prediction error uncertainty, which is supervised by real reward. In contrast, `LinTS` is perturbing the system parameter, when can be wrong if the system modeling is wrong.*

### 3.3 EFFICIENT IMPLEMENTATION

By the attractive computational properties of Gaussian distribution, the computational cost of `LinReBoot` can be reduced significantly when Gaussian Bootstrap weights are generated. Formally: assume $\omega_{k,t,i} \sim N(0, \sigma_\omega^2), \forall k, t, i$, recalling (7), for $k \in [K]$ and any $t \geq 1$, bootstrapped mean $\tilde{\mu}_{k,t}$ follows a Gaussian distribution,

$$\tilde{\mu}_{k,t} | \mathcal{F}_{t-1} \sim N(\hat{\mu}_{k,t}, \sigma_\omega^2 s_{k,t-1}^{-2} RSS_{k,t}). \qquad (9)$$

Such Gaussian-distributed property of $\tilde{\mu}_{k,t}$ indicates that if we can update $\hat{\mu}_{k,t}$, $s_{k,t-1}$ and $RSS_{k,t}$ incrementally for arm $k$, this bootstrapped mean $\tilde{\mu}_{k,t}$ can be generated by Gaussian generator without inner loop for generating weights. The first two terms, $\hat{\mu}_{k,t}$ and $s_{k,t-1}$, are naturally updated in incremental manner. For $RSS_{k,t}$, following decomposition ensures an incremental update,

$$RSS_{k,t} = \sum_{i=1}^{s_{k,t-1}} r_{k,i}^2 + s_{k,t-1} \hat{\mu}_{k,t}^2 - 2\hat{\mu}_{k,t} \sum_{i=1}^{s_{k,t-1}} r_{k,i}.$$

Then an efficient generation for $\tilde{\mu}_{k,t} | \mathcal{F}_{t-1}$ is ensured by the incremental updates for $\hat{\mu}_{k,t}$, $s_{k,t-1}$, $\sum_{i=1}^{s_{k,t-1}} r_{k,i}^2$, $\sum_{i=1}^{s_{k,t-1}} r_{k,i}$. Furthermore, since the residual bootstrap weights are generated independently, $\tilde{\mu}_{k,t}$ among arms are also independent given historical randomness and can be sampled from one multivariate Gaussian generation simultaneously. Formally, $\tilde{\boldsymbol{\mu}}^{(t)} = (\tilde{\mu}_{1,t}, \ldots, \tilde{\mu}_{K,t})^\top$ is conditional distributed as

$$\tilde{\boldsymbol{\mu}}^{(t)} | \mathcal{F}_{t-1} \sim N_K(\hat{\boldsymbol{\mu}}^{(t)}, \boldsymbol{\Sigma}_\omega^{(t)}), \qquad (10)$$

where $\hat{\boldsymbol{\mu}}^{(t)} = (\hat{\mu}_{1,t}, \ldots, \hat{\mu}_{K,t})^\top$ and $\boldsymbol{\Sigma}_\omega^{(t)}$ is a diagonal matrix with diagonal elements $\sigma_\omega^2 s_{k,t-1}^{-2} RSS_{k,t}$. Detailed steps and more illustration about efficient implementation is provided in Appendix D.7.1. Moreover, an empirical study about computational efficiency is conducted in Appendix D.7.2 and Table.3 provides the computational cost of our proposed `LinReBoot` as well as other baseline algorithms.

## 4 OPTIMISM DESIGN

**Optimistic Estimated Discrepancy.** This section identifies and demystifies the technical challenge of implementing `ReBoot` principle in the stochastic linear bandit problem. The key is to conduct a detailed investigation to produce probabilistic control on the behavior of the 'Optimistic Estimate Discrepancy (OED)' of the `LinReBoot` policy (8). In principle, the **OED** is given by

$$\textbf{OED} = \text{Optimism} \times \texttt{Action Context Norm}, \quad (11)$$

where the `Action Context Norm` is given by $\|\boldsymbol{x}_k\|_{\boldsymbol{V}_t^{-1}}$ and Optimism is given by $c_{t,k}$ for the $k$th action at time $t$, defined in (14). Design of $c_{t,k}$ will be elaborated in Section 4.1.

**Sufficient Explored Arms.** We define the concept of *Sufficient Explore Arms* to facilitate the formal regret analysis of `LinReBoot`. Intuitively, an arm is *sufficient explored* if its index produced by the policy (8) is less than the mean reward of the optimal arm. Technically, we say an arm $k$ is *sufficiently explored* at time $t$ if the adopted OED $(c_{t,k}\|\boldsymbol{x}_k\|_{\boldsymbol{V}_t^{-1}})$ is bounded by its optimal gap $(\Delta_k)$.

The above notion of sufficient explored arm defines the concept of "set of sufficient explored arms" $\mathcal{S}_t$, formally

$$\mathcal{S}_t := \{k \in [K] : c_{t,k}\|\boldsymbol{x}_k\|_{\boldsymbol{V}_t^{-1}} < \Delta_k\}, \qquad (12)$$

where and $c_{t,k}$ is the collaborated optimism and $c_{t,k}\|\boldsymbol{x}_k\|_{\boldsymbol{V}_t^{-1}}$ is an optimistic estimate of discrepancy of policy index (8).

The key consequence of set (12) is that, any member in $\mathcal{S}_t$ enjoys the property

$$\forall j \in \mathcal{S}_t \cap [K] : \tilde{\mu}_{j,t} < \mu_1; \qquad (13)$$

that is, the `LinReBoot` policy always avoids an index (8) from sufficiently explored subset such that the bootstrapped mean of this index is less than the optimal mean reward unless all arm are sufficiently explored. (see equation (82) in the proof of Lemma A.1 at section B.1 for technical details).

## 4.1 COLLABORATE OPTIMISM

Here we elaborate on the collaborated optimism adopted in the definition of sufficient explored arms (12). Concretely, the collaborated optimism has a form

$$c_{t,k} = c_1(t,k) + c_2(t,k), \qquad (14)$$

where $c_1(t,k)$ is called *sample optimism* and $c_2(t,k)$ is called *bootstrap optimism* for arm $k$ at time $t$.

**Sample Optimism.** The sample optimism $c_1(t,k)$ serves as a control on the event that "the realized sample estimate discrepancy (ED) is bounded by sample OED":

$$E_{t,k} := \{|\hat{\mu}_{k,t} - \mu_k| \leq c_1(t,k)\|\boldsymbol{x}_k\|_{\boldsymbol{V}_t^{-1}}, \} \qquad (15a)$$

$$E_t := \bigcap_{k=1}^{K} E_{t,k}, \qquad (15b)$$

where $c_1(t,k)$ is a constant which can be tuned by our `LinReBoot` algorithm, making the bad event $\bar{E}_{t,k}$ and $\bar{E}$ become unlikely. In fact, this $E_{t,k}$ is the event that the least squared estimation is "close" to the true mean reward for arm $k$ at round $t$. In section 5, the probability of the bad event $\bar{E}_t$ is controlled by a parameter tuned by users based on lemma 5.1.

**Bootstrap Optimism.**

The bootstrap optimism $c_2(t,k)$ serves as a control on the event that "the realized bootstrap ED is bounded by bootstrap OED":

$$E'_{t,k} := \{|\tilde{\mu}_{k,t} - \hat{\mu}_{k,t}| \leq c_2(t,k)\|\boldsymbol{x}_k\|_{\boldsymbol{V}_t^{-1}}\}, \qquad (16a)$$

$$E'_t := \bigcap_{k=1}^{K} E'_{t,k}, \qquad (16b)$$

where $c_2(t,k)$ is also a constant controlling the conditional probability of the bad event $\bar{E}'_t$. This $c_2(t,k)$ can be tuned by our `LinReBoot` algorithm as well. Similar to $E_{t,k}$, this $E'_{t,k}$ is the event that the residual bootstrap based estimation is "close" to the least squared estimate $\hat{\mu}_{k,t}$ for arm $k$ at round $t$. In section 5, the probability of bad event $\bar{E}'_t$ is controlled by a parameter tuned by users based on lemma 5.2.

## 4.2 OPTIMISM DESIGN

**Choice of sample optimism ($\alpha$).** The goal of this part is to illustrate how to pick the sample OED such that the event (15) holds with probability at least $1 - \alpha$ for a given confidence budget $\alpha \in (0,1)$. Formally, the goal is to find a sample OED function $c_1(t,k) :$

$[n] \times [K] \mapsto \mathbb{R}$ such that the event (15a) holds with probability at least $1 - \alpha_k$. To meet the purpose of the risk control, we specify the sample OED function with form

$$c_1(t,k) := R_2\sqrt{d\log((1 + tL^2/\lambda)/\alpha_k)} + \lambda^{1/2}S_2. \quad (17)$$

Lemma 5.1 gives the formal result on why such choice has confidence budget at most $\alpha_k$. For regret analysis, define $\alpha_{\min} = \min_{k \in [K]} \alpha_k$ and $\boldsymbol{\alpha} = (\alpha_1, ..., \alpha_K)^\top$.

**Choice of bootstrap optimism ($\beta$).** The goal of this part is to pick bootstrapped OED such that the event (16) holds with probability at least $1 - \beta$ for given confidence budget $\beta \in (0,1)$. Formally, the goal is to find a sample OED function $c_2(t,k) : [n] \times [K] \mapsto \mathbb{R}$ such that the event (16a) holds with probability at least $1 - \beta_k$. To meet the purpose of the risk control, we specify the bootstrapped OED function with form

$$c_2(t,k) := \sqrt{(2\sigma_\omega^2 RSS_{k,t}\log(2/\beta_k))/s_{k,t-1}^2}\|\boldsymbol{x}_k\|_{\boldsymbol{V}_t^{-1}}^2. \quad (18)$$

Lemma 5.2 gives the formal result on why such choice has a confidence budget at most $\beta_k$. For regret analysis, let $\beta_{\min}$ be the smallest $\beta_k$, $\forall k \in [K]$ and $\boldsymbol{\beta} = (\beta_1, ..., \beta_K)^\top$.

## 4.3 OPTIMISM FOR OPTIMAL ARM

**Sample-Bootstrap OED ratio of the optimal arm (b).** Indicated by the regret analysis in [Kveton et al., 2020a], instead of controlling the exploration independently, the relation between two sources of explorations needs to be considered because this relation is critical for finding the optimal action. To meet such observation, we define a good event,

$$E''_t := \{\tilde{\mu}_{1,t} - \hat{\mu}_{1,t} > c_1(t,1)\|\boldsymbol{x}_1\|_{\boldsymbol{V}_t^{-1}}\}. \qquad (19)$$

Given the good event $E''_t$, the policy index $\tilde{\mu}_{1,t}$ of the optimal arm enjoys further positive bias, hence the agent will have better chance to make optimal action.

In particular, we highlight a constant $b$ used to measure the ratio of the sample optimism (17) to the bootstrap optimism (18); formally, we require $b$ satisfies

$$c_1(t,1)/c_2(t,1) \geq b \cdot \sqrt{2\log(2/\beta_1)}. \qquad (20)$$

Intuitively, the constant $b$ measures the relation between sample OED and bootstrap OED of the optimal arm. This $b$ plays an important role of the probability lower bound of event (19) (See Lemma 5.3). Note that, if (20) holds, we have the lower bound (26) ; otherwise, we have the lower bound (27). In both cases, we have a lower bound for the event (19).

| Notation | Definition |
|---|---|
| $\zeta_1(n,d)$ | $\left(L_2\sqrt{d\log\left(\frac{1+nL^2/\lambda}{\alpha_{\min}}\right)}+\lambda^{1/2}S_2\right)\times$ $\sqrt{2(n-K)d\log(1+\sum_{i=1}^r\sigma_i^2/d\lambda)}$ |
| $\zeta_2(n,d)$ | $\sqrt{2\sigma_\omega^2 log(\frac{2}{\beta_{\min}})}\times$ $\sqrt{2(n-K)d\log(1+\sum_{i=1}^r\sigma_i^2/d\lambda)}$ |
| $\zeta_3(n)$ | $2K\sqrt{4L_2\sigma_\omega^2\log\left(\frac{2}{\beta_{\min}}\right)}(\log n+1)$ |
| $\zeta_4(n)$ | $2S_2L((n-K)(\alpha+\beta)+K-1)$ |

Table 1: Notations in Regret Analysis

**Good event for optimal arm ($\gamma$).** Here we introduce the event that over exploration and under exploration of the optimal arm have been avoided simultaneously. Formally, the constant $\gamma$ is the probability that the bandit index (8) is not over-exploration (Event $E_t'$) and also not under-exploration (Event $E_t''$)

$$\{c_1(t,1)<(\tilde{\mu}_{1,t}-\hat{\mu}_{1,t})/\|\boldsymbol{x}_1\|_{\boldsymbol{V}_t^{-1}}<c_2(t,1)\}. \quad (21)$$

Technically, we can show that the probability of the event (21) is lower bounded by the term

$$\mathbb{P}_t(E_t'')-\mathbb{P}_t(\bar{E}_t'), \quad (22)$$

with probability at least $1-\gamma$ (Lemma 5.4). Such lower bound is translated into an upper bound in regret analysis.

## 5 FORMAL RESULTS

### 5.1 REGRET BOUND FOR LINREBOOT

**Theorem 5.1.** *Under Assumptions 1, 2, 3 and technical conditions (32) and (74), with probability at least $1-(\delta+\gamma)$, the expected regret of Algorithm 1 is bounded as,*

$$\begin{aligned}R_n \leq &C_1(\alpha_1,\boldsymbol{\beta},\gamma,b)\zeta_1(n,d)\\&+C_2(\boldsymbol{\alpha},\boldsymbol{\beta},\gamma,b,\delta)\zeta_2(n,d)\\&+C_1(\alpha_1,\boldsymbol{\beta},\gamma,b)\zeta_3(n)+\zeta_4(n),\end{aligned} \quad (23)$$

*where $\zeta_1$, $\zeta_2$, $\zeta_3$ and $\zeta_4$ are defined in Table 1 and $C_1$, $C_2$, $M_1$, $M_2$ are described in Table 2.*

*Proof.* See Appendix A.1.

$\square$

**Corollary 5.2.** *Let $\boldsymbol{\alpha}=\boldsymbol{\beta}=\frac{1}{\sqrt{n}}\mathbf{1}$, the order of high probability upper bound in Theorem 5.1 is $\tilde{O}(d\sqrt{n})$.*

*Proof.* See Appendix A.2.

$\square$

Corollary 5.2 shows that our regret bound scales as the regret bound of Linear Thompson sampling [Agrawal and Goyal, 2013b] and Linear PHE [Kveton et al., 2020a].

### 5.2 VALIDATE SAMPLE OPTIMISM

**Lemma 5.1.** *Under Assumptions 1, 2, 3 and choose $c_1(t,k)$ as (17), $\mathbb{P}(\bar{E}_{t,k})$, the probability of bad event corresponded to least squared estimation described in (15), is controlled. Formally, $\forall k\in[K]$, $\forall\alpha_k>0$, $\forall t\geq 1$,*

$$\mathbb{P}(|\hat{\mu}_{k,t}-\mu_k|\leq c_1(t,k)\|\boldsymbol{x}_k\|_{\boldsymbol{V}_t^{-1}})\geq 1-\alpha_k. \quad (24)$$

*Consequently, we have $\mathbb{P}(\bar{E}_t)\leq\alpha:=\sum_{k=1}^K\alpha_k$.*

*Proof.* See Appendix A.3.

$\square$

Lemma 5.1 supports that the choice of $c_1(t,k)$ at (17) for the sample optimism event (15) is valid with confidence budget $\alpha$.

### 5.3 VALIDATE BOOTSTRAP OPTIMISM

**Lemma 5.2.** *Suppose bootstrap weights are Gaussian. Pick $c_2(t,k)$ as (18). The conditional probability of bad event corresponding to residual bootstrap exploration described in (16), $\mathbb{P}_t(\bar{E}_{t,k}')$, is controlled. Formally, $\forall k\in[K]$, $\forall\beta_k>0$, $\forall t\geq 1$*

$$\mathbb{P}_t(|\tilde{\mu}_{k,t}-\hat{\mu}_{k,t}|\leq c_2(t,k)\|\boldsymbol{x}_k\|_{\boldsymbol{V}_t^{-1}})\geq 1-\beta_k. \quad (25)$$

*Consequently, we have $\mathbb{P}_t(\bar{E}_t')\leq\beta:=\sum_{k=1}^K\beta_k$.*

*Proof.* See Appendix A.4.

$\square$

Lemma 5.2 supports that the choice of $c_2(t,k)$ at (18) for the sample optimism event (16) is valid with confidence budget $\beta$.

### 5.4 SAMPLE-BOOTSTRAP RATIO

**Lemma 5.3.** *Under Assumptions 1, 2, 3. Suppose bootstrap weights are Gaussian. The conditional probability of anti-concentration for optimal arm described in*

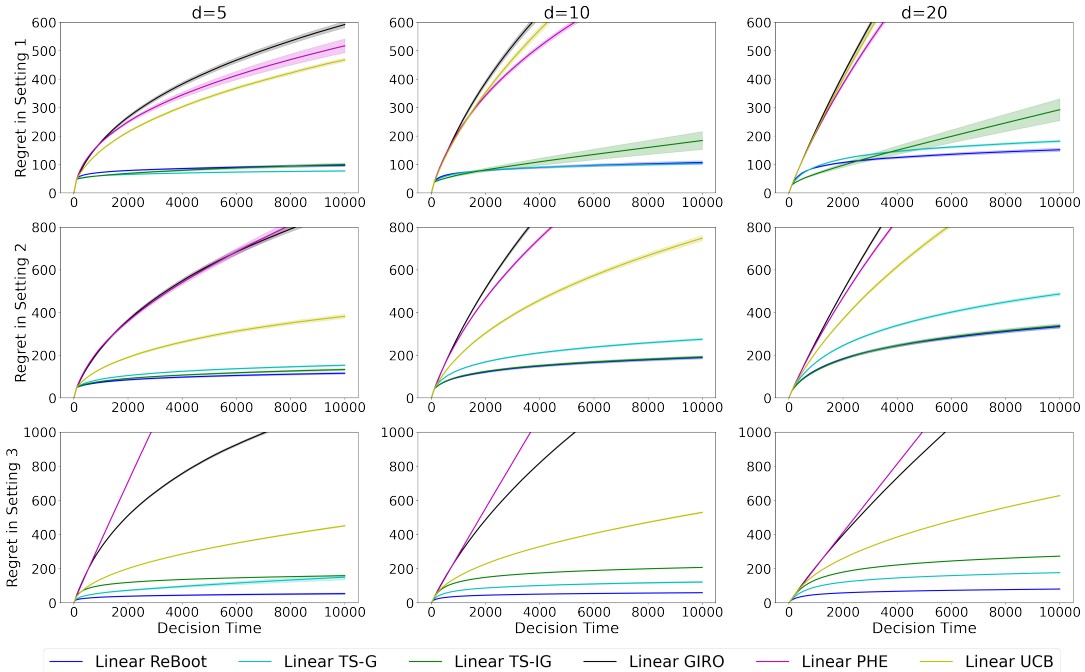

Figure 1: Comparison of `LinReBoot` with Gaussian Bootstrap weights to baselines under three linear bandit problems and three different context dimension $d$. First row referred to the setting in Section 6.1, second row is for Section 6.2 and the last row is for Section 6.3. Three columns refer to $d = 5$, $d = 10$ and $d = 20$ respectively.

(19), $\mathbb{P}_t(\bar{E}_t'')$, has lower bound. Formally, if b satisfies (20),

$$\mathbb{P}_t(E_t'') \geq \frac{b}{\sqrt{2\pi}} \exp\left(-\frac{3c_1^2(t,1)s_{1,t-1}^2\|\boldsymbol{x}_1\|_{\boldsymbol{V}_t^{-1}}^2}{2\sigma_\omega^2 RSS_{1,t}}\right). \tag{26}$$

Otherwise,

$$\mathbb{P}_t(E_t'') \geq \Phi(-b), \tag{27}$$

where $\Phi$ is the CDF of standard normal distribution.

*Proof.* See Appendix A.5.

□

Lemma 5.3 provides the lower bound result for good event $E_t''$. The result indicates that, if the bootstrap optimism is not 'too large', then the `LinReBoot` procedure can enjoy additional regret reduction.

## 5.5 VALIDATE GOOD EVENT

**Lemma 5.4.** *Under Assumptions 1, 2, 3 and suppose Bootstrap weights are Gaussian. Assume b satisfies a technical condition (74). Then, with probability at least*

$1 - \gamma$, $\mathbb{P}_t(E_t'') - \mathbb{P}_t(\bar{E}_t')$ *has lower bound,*

$$\frac{b}{\sqrt{2\pi}} \exp\left(-\frac{3s_{1,t-1}^{3/2}c_1^2(t,1)\|\boldsymbol{x}_1\|_2^2}{8\sigma_\omega^2(\sigma_{\min}^2+\lambda)\sqrt{\frac{1}{M_2}\log\left(\frac{M_1}{1-\gamma}\right)}}\right) - \beta, \tag{28}$$

*where $M_1$ and $M_2$ are defined in Table 2.*

*Proof.* See Appendix A.6.

□

Lemma 5.4 provided the a high probability lower bound for the difference between probability of the event for anti-concentration $E_t''$ and probability of bad event discussed in bootstrap optimism in Section 4.1. This lower bound is also for probability of 'not under and not over exploration' event (21). Lemma 5.4 links the sample optimism and bootstrap optimism and holds a right amount of exploration of the optimal arm.

## 6 EXPERIMENTS

In this section, we conduct empirical studies under three settings: Stochastic Linear Bandit, Contextual

Linear Bandit and Linear Bandit with Covariates. Our `LinReBoot` is compared to several baselines including `LinTS-G` [Agrawal and Goyal, 2013b, Lattimore and Szepesvári, 2020], `LinTS-IG` [Honda and Takemura, 2014, Riquelme et al., 2018], `LinPHE` [Kveton et al., 2020a], `LinGIRO` [Kveton et al., 2019b] and `LinUCB` [Abbasi-Yadkori et al., 2011, Lattimore and Szepesvári, 2020] . More details about baselines can be found in Appendix D.6.

## 6.1 STOCHASTIC LINEAR BANDIT

We compare `LinReBoot` to other linear bandit algorithms under stochastic linear bandit described in Section 2. We experiment with several dimensions $d$ including 5, 10 and 20. $K$ is chosen as 100. Synthetic data generation for this setting is deferred to Appendix D.2 in the supplementary material. **Results.** The first row of Figure 1 reports the results for Stochastic Linear Bandit setting. Our `LinReBoot` rivals `LinTS-G` and `LinTS-IG` while substantially exceeds `LinGIRO`, `LinPHE` and `LinUCB`. When $d$ increases, the performance of `LinReBoot` rivals and exceeds the best of other methods.

## 6.2 CONTEXTUAL LINEAR BANDIT

In the second experiment, we compare `LinReBoot` to other linear bandit algorithms under Contextual Linear Bandit where the contexts are generated from some distributions by arms. Note that this setting matches previous work [Chu et al., 2011]. Linear bandit algorithms can also be applied under this kind of environment. In our experiment, the `LinReBoot` is implemented as Algorithm 2 in Appendix D.1. Like the setting in Section 6.1, the dimension of $d$ is chosen as 5 or 10 or 20 and the synthetic data generation for this setting is described in Appendix D.2. **Results.** The second row of Figure 1 reports the results for Contextual Linear Bandit. Our `LinReBoot` rival `LinTS-G` and substantially exceed `LinTS-IG`, `LinGIRO`, `LinPHE` and `LinUCB`. When $d$ increases, the performance of `LinReBoot` rivals `LinTS-IG` and exceeds others.

## 6.3 BANDIT WITH COVARIATES

Our last experiment is conducted under the setting of linear bandit with covariates, which is also called linear parametrized bandit by [Rusmevichientong and Tsitsiklis, 2010] . This problem is significantly different from the previous two problems in the following ways. Each arm has its true parameter $\boldsymbol{\theta}_k$. That is, each arm has its estimate $\hat{\boldsymbol{\theta}}_k$ from the ridge regression procedure in Section 3.2. Also, unlike the setting in Section 6.2, the contexts are generated from a distribution that is independent of arms. Thus the overall task in this setting is not only the estimation of the target parameter $\boldsymbol{\theta}$, but also the detection of which arm a context belongs to. This case is also referred to as the online decision-making under covariates [Bastani and Bayati, 2020]. For the `LinReBoot` in this setting, detailed algorithm is provided as Algorithm 3 in Appendix D.1. $d$ is chosen as 5 or 10 or 20 and $K = 10$. Synthetic data generation for this setting is described in Appendix D.2. **Results.** The third row of Figure 1 reports the results for Linear Bandit with Covariates. Our `LinReBoot` exceeds all competing algorithms `LinTS-G`, `LinTS-IG`, `LinGIRO`, `LinPHE` and `LinUCB`.

**Summary.** From Figure 1, the proposed `LinReBoot` is always the top 3 algorithms under all settings and all choice of dimension $d$. More specifically, `LinReBoot` is clearly comparable to the state-of-the-art Linear Thompson Sampling algorithms(`LinTS-G`, `LinTS-IG`) or even outperforms them in many cases. Regarding the computational cost, from Table.3, our proposed `LinReBoot` is consistently computational efficient among all settings compared to `LinTS-G`, `LinTS-IG` and `LinUCB` under all three settings.

## 7 CONCLUSION

We propose `LinReBoot` algorithm for stochastic linear bandit problems. In theory, we prove `LinReBoot` that secures $\tilde{O}(d\sqrt{n})$ high probability expected regret. Empirically, we show `LinReBoot` rivals `LinTS-G`, `LinTS-IG` and exceeds `LinPHE`, `LinGIRO` and `LinUCB`, which supports the easy-generalizability of `ReBoot` principle in [Wang et al., 2020] under various contextual bandit settings including Stochastic Linear Bandit, Contextual Linear Bandit, and Linear Bandit with Covariates.

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
