# OpenReview forum: "Residual Bootstrap Exploration for Stochastic Linear Bandit"
_auai.org/UAI/2022/Conference — UAI 2022 Poster_

### Official Review · Reviewer_dEFC · 2022-04-06

**Q2(1) Originality/Novelty:** 3
**Q2(2) Significance/Impact:** 2
**Q2(3) Correctness/Technical Quality:** 2
**Q2(6) Clarity Of Writing:** 3
**Q6 Overall Score:** 6
**Q8 Confidence In Your Score:** 1

**Q1 Summary And Contributions:**

This paper proposes a residual bootstrapping algorithm for stochastic linear bandits, and shows that it achieves sublinear regret with high probability.

**Q2 Assessment Of The Paper:**

More detailed information regarding each of these aspects is given below:

**Q2(4) Quality Of Experiments (Optional):**

3: Good: The experimental evaluation is adequate, and the results convincingly support the main claims.

**Q2(5) Reproducibility:**

3: Good: Key resources (e.g., proofs, code, data) are available and key details (e.g., proofs, experimental setup) are sufficiently well-described for competent researchers to confidently reproduce the main results.

**Q3 Main Strengths:**

The algorithm looks interesting, and I could see the contribution of this paper being important both theoretically and in practice.

**Q4 Main Weakness:**

-

**Q5 Detailed Comments To The Authors:**

page 2, paragraph "Regret": the assumption that arm 1 is the *unique* optimal arm is not without loss of generality.
page 3, Assumption 1: The x_k vectors (finitely many) are known ahead of time, right? Then what does it mean to say they are bounded?

The grammar of could be much improved, though in most cases it didn't hinder under of the paper. Below are some cases where it did:
page 1 "arms information" -> "arm's information"
page 1 "collaborate": I think you mean a different word here
page 3, "Actions Exploring": "subroutines" -> "subroutine"; "sample" -> "samples"; "output" -> "outputs"

**Q7 Justification For Your Score:**

I am not familiar at all with relevant parts of the bandit literature, so my assessment of all aspects of this paper except quality of writing is an educated guess.

**Q9 Complying With Reviewing Instructions:**

1: Yes.

---

### Official Review · Reviewer_iNc2 · 2022-04-09

**Q2(1) Originality/Novelty:** 3
**Q2(2) Significance/Impact:** 3
**Q2(3) Correctness/Technical Quality:** 3
**Q2(6) Clarity Of Writing:** 2
**Q6 Overall Score:** 6
**Q8 Confidence In Your Score:** 3

**Q1 Summary And Contributions:**

This paper studies the stochastic linear bandit (SLB) problem and proposes a bonus based on residual bootstrap for exploration. The algorithm is simple and intuitive. The main contribution of this paper is to provide an \tilde{O}(d\sqrt{n}) bound for the bootstrap-based method.

**Q10 Ethical Concerns (Optional):**

No ethical concerns.

**Q2 Assessment Of The Paper:**

More detailed information regarding each of these aspects is given below:

**Q2(4) Quality Of Experiments (Optional):**

3: Good: The experimental evaluation is adequate, and the results convincingly support the main claims.

**Q2(5) Reproducibility:**

3: Good: Key resources (e.g., proofs, code, data) are available and key details (e.g., proofs, experimental setup) are sufficiently well-described for competent researchers to confidently reproduce the main results.

**Q3 Main Strengths:**

The main strength of the paper is as follows:

- this paper provides a new algorithm for stochastic linear bandit based on bootstrap. The algorithm is easy to implement and can be extended to various types of linear bandit problems.

- this paper provides an $O(d\sqrt{T})$ regret bound for the SLB problem, which is novel for bootstrap-based algorithms and matches the results of other methods in literature (like linUCB and LinTS)

- experiments on various problem setups are conducted to validate the effectiveness of the proposed method.

**Q4 Main Weakness:**

The main weakness of the paper is as follows:

- About the assumption: The paper has imposed several additional assumptions and technical assumptions to obtain the $\tilde{O}(d\sqrt{T})$ bound for the bootstrap-based algorithm than other methods in the literature. Although some of them are standard for SLB problems, like Assumption 1 and the sub-Gaussian noise, others are hardly seen previously.

- About the clarity: this first point on clarity is related to the assumption issue, as some technical conditions are hard to understand for me. For example, it is unclear what is the intuition behind the condition (74) and how hard this condition could be satisfied.



**Q5 Detailed Comments To The Authors:**

This paper provides a bootstrap-based algorithm for the SLB problem and provides an $O(d\sqrt{T})$ regret bound for it. Although the bound for the bootstrap-based algorithm is novel to me, I still have some concerns about the assumptions and clarity of this paper.

As mentioned in Q4, many assumptions and technical conditions are introduced in this paper. Some of the assumptions are hard to understand for me, like the condition (74). I think it would be better if the author could provide some illustration for the assumption. Although it might be hard, the author should at least discuss how hard this condition can be satisfied.

Besides, it is strange to me that an assumption on the lower bound of the noise $\epsilon$ is imposed in this paper. As shown in table 2, the lower bound L1 appears in the denominator inside the exponential function. It seems that if there is no noise in some iterations, the constant M1 will come unbounded.

Meanwhile, I think it would be better for the author to highlight their technical contribution. Although the result for SLB problem is novel, the bootstrap-baed method has received many studies in the MAB case. The paper would have more contribution if the author could highlight the main technical challenge to extend the analysis for MAB to the SLB case.

I would raise my score if the author could address the above concerns on the assumptions and the novelty.

**Q7 Justification For Your Score:**

The main strength of this paper is to propose a simple and easy-to-implement bootstrap-based algorithm for the SLB problem. The experiments show that the proposed method enjoys comparable empirical performance to other methods in the literature and is even superior in the high dimensional case. Although the regret bound of the proposed is established on several assumptions (see weakness in Q4), I still find the merit of this paper overrides the flaw and tend to accept this paper.


**Q9 Complying With Reviewing Instructions:**

1: Yes.

---

### Official Review · Reviewer_1Cn4 · 2022-04-11

**Q2(1) Originality/Novelty:** 3
**Q2(2) Significance/Impact:** 2
**Q2(3) Correctness/Technical Quality:** 3
**Q2(6) Clarity Of Writing:** 2
**Q6 Overall Score:** 4
**Q8 Confidence In Your Score:** 2

**Q1 Summary And Contributions:**

This paper considers stochastic linear bandit problems and proposes a residual bootstrap-based exploration algorithm called LinReBoot, which seems to be inspired by prior work including an algorithm known as ReBoot for multi-armed bandit problems. One of the innovations is a regret bound, that matches bounds for linear Thompson sampling algorithms. Through experiments with synthetic data, the proposed algorithm is shown to be comparable or better than some baselines.

**Q2 Assessment Of The Paper:**

More detailed information regarding each of these aspects is given below:

**Q2(4) Quality Of Experiments (Optional):**

2: Fair: The experimental evaluation is weak: important baselines are missing, or the results do not adequately support the main claims.

**Q2(5) Reproducibility:**

2: Fair: Key resources (e.g., proofs, code, data) are unavailable but key details (e.g., proof sketches, experimental setup) are sufficiently well-described for an expert to confidently reproduce the main results.

**Q3 Main Strengths:**

1.	The work undertaken to prove the theoretical guarantee for the algorithm seems like the most important innovation in the paper.
2.	The results based purely on synthetic data seem promising.
3.	There is a claim in the experiments section that the ideas are applicable to variations in the underlying problem. If this is true, this would expand the scope of the proposed approach.


**Q4 Main Weakness:**

1.	For me, the main weakness of the paper is the legibility. I had a very difficult time following the paper, and in fact there is a large portion of the paper that I could not follow. A part of the reason for this is that I am not aware of the closest literature, but another important part is the writing style of the paper. The results are mentioned with not a lot of intuitive explanation, and there are several places in the paper where ideas are mentioned before they are actually described. There are also numerous typos and grammatical errors. Due to my general lack of understanding, it is hard for me to verify most of the results and to evaluate the work effectively.
2.	There is no mention in the main text about how the synthetic data is generated for experiments. This makes it hard to understand if the generation is reasonable and fair for all methods. Furthermore, there is no experiment that uses real data.


**Q5 Detailed Comments To The Authors:**

I recommend writing the Abstract so that it is more easily accessible to a general reader. For instance, explain what “d” and “n” mean in the regret bound. I also recommend simplifying some of the literature discussion in Section 1. It would definitely not be clear enough for a generic reader.

The phrase “demystify the bootstrap optimism” is used in several places in the paper, but the paper itself did nothing to demystify this for me. If anything, it only mystified matters as I was not aware of any mystery previously!

What is “easy generalizable” on p2?

The filtration F_t could use a better explanation on p2.

The paragraph before “Notation” on p2 is a bit unclear. From what I can tell, the most important related approaches -- ReBoot, GIRO and PHE -- are not explained clearly enough.

“Boundness” is misspelled in the body of Assumption 1. In general, there are numerous typos – I stopped identifying them at some point so hopefully the authors can find them and fix these themselves.

The beginning quotation marks (both for single and double quotations) are incorrectly specified throughout several places in the paper, ex: “sample-bootstrap …” on p1.

I found some of the summaries (like on p4) to be quite helpful.

P5 introduces several terms that were 1) named in multiple ways (are these typos?), ex: “sufficient explore arms” and “sufficient explored arms”. I don’t understand what any of them really mean. Also, I found the naming to be less than ideal, for instance, should it be “sufficiently explored arms” in this example?

“Good” and “bad” events are described without much explanation. Again, I don’t think these are the most descriptive names.

Section 4.2 is an example where references are made to later in the paper. When perhaps this should have been combined with the content in Section 5.

While describing the results on p7, I think the authors should mention the axes for graphs in Fig 1.

**Q7 Justification For Your Score:**

From what I can tell, this paper seems to make useful contributions to stochastic linear bandits. However, I find the legibility issue in the current version to be severe enough to outweigh the work’s strengths. Please note that my confidence in this assessment is low and I will therefore rely on more knowledgeable reviewers in the discussion phase.

**Q9 Complying With Reviewing Instructions:**

1: Yes.

---

### Official Review · Reviewer_Q6PB · 2022-04-13

**Q2(1) Originality/Novelty:** 3
**Q2(2) Significance/Impact:** 3
**Q2(3) Correctness/Technical Quality:** 4
**Q2(6) Clarity Of Writing:** 4
**Q6 Overall Score:** 7
**Q8 Confidence In Your Score:** 3

**Q1 Summary And Contributions:**

This paper proposes the use of residual bootstraps for contextual linear bandits. The authors show regret of results of ~dsqrt(n) and provide empirical evidence of the efficacy of the proposed model. Experiments show strong empirical performance.

**Q2 Assessment Of The Paper:**

More detailed information regarding each of these aspects is given below:

**Q2(4) Quality Of Experiments (Optional):**

3: Good: The experimental evaluation is adequate, and the results convincingly support the main claims.

**Q2(5) Reproducibility:**

3: Good: Key resources (e.g., proofs, code, data) are available and key details (e.g., proofs, experimental setup) are sufficiently well-described for competent researchers to confidently reproduce the main results.

**Q3 Main Strengths:**

1. Very interesting problem setting
2. Nice theoretical characterization
3. Well written and organized

**Q4 Main Weakness:**

1. residual bootstrap may be computationally demanding in practice

**Q5 Detailed Comments To The Authors:**

I enjoyed this paper: well motivated, well written, and thorough. The idea of using the residual (wild) bootstrap is a natural and nice one, and the authors do a nice job of presenting the method and providing thorough analysis. I found the experimental analysis to be reasonable, and provide what I interpret to be very compelling results.

One question I have is around Assumption A2: is it the case that we are assuming independent noise here? My understanding for the residual bootstrap is that is the case, though one could extend to non-iid instances using the dependent wild bootstrap of Shao. It would be nice to make this plain in the text.

**Q7 Justification For Your Score:**

Well motivated, strong accompanying analysis and empirical results. Practically relevant to practitioners.

**Q9 Complying With Reviewing Instructions:**

1: Yes.

---

### Decision · Program_Chairs · 2022-05-15

**Decision:**

Accept (Poster)

**Comment:**

Meta Review: This paper studies residual bootstrap exploration in a stochastic linear bandit. The promise of the approach is that it is more general than the traditional approaches to exploration. Although I would not expect a regret analysis beyond linear / GLM bandits, the authors could experiment with non-linear problems, such as in some cited papers (Giro). The reviewers liked the generality of the approach. The main concern seemed to be that the paper is geared towards experts and not a general bandit reader. I strongly recommend that the authors make the paper more accessible, which will expand its reading base. I also looked at the paper and suggest that the authors check all references. Many arxiv papers have been accepted at conferences and should be cited as such.